# Nicotinamide Attenuates the Progression of Renal Failure in a Mouse Model of Adenine-Induced Chronic Kidney Disease

**DOI:** 10.3390/toxins13010050

**Published:** 2021-01-11

**Authors:** Satoshi Kumakura, Emiko Sato, Akiyo Sekimoto, Yamato Hashizume, Shu Yamakage, Mariko Miyazaki, Sadayoshi Ito, Hideo Harigae, Nobuyuki Takahashi

**Affiliations:** 1Division of Nephrology, Endocrinology and Vascular Medicine, Tohoku University Graduate School of Medicine, Sendai 980-8574, Japan; satoshi.kumakura@med.tohoku.ac.jp (S.K.); take-seki@med.tohoku.ac.jp (A.S.); yamashu618@gmail.com (S.Y.); mamiyaza@med.tohoku.ac.jp (M.M.); db554@med.tohoku.ac.jp (S.I.); harigae@med.tohoku.ac.jp (H.H.); nobuyuki.takahashi.a8@tohoku.ac.jp (N.T.); 2Division of Clinical Pharmacology and Therapeutics, Tohoku University Graduate School of Pharmaceutical Sciences, Sendai 980-8578, Japan; giotto.vongola44586yt@gmail.com; 3Department of Medicine, Katta General Hospital, Shiroishi 989-0231, Japan

**Keywords:** nicotinamide, CKD, NAD^+^, adenine-induced CKD model, glycolysis, Krebs cycle

## Abstract

Nicotinamide adenine dinucleotide (NAD^+^) supplies energy for deoxidation and anti-inflammatory reactions fostering the production of adenosine triphosphate (ATP). The kidney is an essential regulator of body fluids through the excretion of numerous metabolites. Chronic kidney disease (CKD) leads to the accumulation of uremic toxins, which induces chronic inflammation. In this study, the role of NAD^+^ in kidney disease was investigated through the supplementation of nicotinamide (Nam), a precursor of NAD^+^, to an adenine-induced CKD mouse model. Nam supplementation reduced kidney inflammation and fibrosis and, therefore, prevented the progression of kidney disease. Notably, Nam supplementation also attenuated the accumulation of glycolysis and Krebs cycle metabolites that occurs in renal failure. These effects were due to increased NAD^+^ supply, which accelerated NAD^+^-consuming metabolic pathways. Our study suggests that Nam administration may be a novel therapeutic approach for CKD prevention.

## 1. Introduction

Chronic kidney disease (CKD) is a worldwide burden affecting approximately 10% of the global population [1,2,3]. Patients with CKD have a high risk of cardiovascular events [4,5], infection [6,7], and malignancy [8,9]. Moreover, CKD is an irreversible disease as kidney function cannot be restored unless the patients undergo renal replacement therapy such as dialysis or kidney transplantation. Thus, patients with CKD face high medical expenses [10], low quality of life [11,12], and high mortality rate [3]. Despite many studies have focused on CKD treatment worldwide, no therapeutic breakthroughs have yet emerged.

CKD is known to affect multiple homeostasis such as energy balance, immune systems, and fluid homeostasis [13]. With respect to energy metabolism, patients with CKD exhibit changes in glycolysis [14], fatty acid oxidations [15], and the Krebs cycle [16]. All the above-mentioned metabolic systems produce ATP and, therefore, provide the energy supply to multiple organs, including kidneys. Nicotinamide adenine dinucleotide (NAD^+^), a co-enzyme in many metabolic reactions, is required in several steps of ATP-producing pathways. NAD^+^ is implicated in neurodegeneration [17], insulin secretion [18], and skeletal muscle function [19]. Recent studies have shown that NAD^+^ plays an essential role in preventing acute kidney diseases [20,21]. There is a successful clinical trials administration of nicotinamide (Nam), a precursor of NAD^+^, on perioperative patients, and showed that Nam is effective on preventing perioperative acute renal failure [22]. However, few studies have addressed the therapeutic relevance of NAD^+^ in CKD [23].

Different metabolic pathways control NAD^+^ synthesis and degradation [24]. In particular, the Preiss–Handler pathway is responsible for NAD^+^ synthesis from nicotinic acid, while a de novo biosynthetic pathway results in NAD^+^ production from tryptophan. These two pathways synthesize NAD^+^ from diet intake. A distinct mechanism, the salvage pathway, results in NAD^+^ synthesis from nicotinamide (Nam), nicotinamide riboside (NR), and nicotinamide mononucleotide (NMN), which derive from NAD^+^ metabolism. Several studies have explored the effects of salvage pathway stimulation by the administration of Nam, NR, and NMN. These investigations have focused on the activation of sirtuins induced by increased NAD^+^ levels. Sirtuins are DNA deacetylating enzymes that regulate gene expression, and have been reported to affect DNA repair, metabolism, and inflammation in various organs [18,24].

We have previously described the effects of Nam in several disease models. Nam attenuates preeclampsia-related kidney injury in a mouse model of disease by lowering the blood pressure, resulting in improved fetal growth [25]. In addition, we found that Nam decreases urinary protein excretion in MRL/lpr mice prone to lupus nephritis [26]. Thus, Nam plays a renoprotective role in a mouse model of kidney disease.

However, the therapeutic relevance of Nam in CKD is not well established. To address this issue, we explored the effects of Nam in an adenine-induced CKD mouse model by using a metabolomics approach.

## 2. Results

### 2.1. Prophylactic Nam Supplementation Prevented the Progression of Kidney Disease

In this study, mice were fed an adenine-enriched or a normal diet. Some animals in each group received Nam at three different concentrations (0.3%, 0.6%, and 1.2%) in drinking water. Thus, the following groups were established: Control diet without Nam (Cont), control diet plus Nam (Nam), adenine-rich diet without Nam (RF), and adenine-rich diet plus Nam (RF + Nam) over a period of six weeks (Figure 1a). In the study term, RF mice had lost food intake and lost weight (Appendix A
Appendix A, Figure 1b). RF + Nam mice showed a recovery of body weight after treatment with 0.6% and 1.2% Nam. RF mice and RF + Nam mice consumed more drinking water compared to Cont mice and Nam mice (Appendix A), which led to the difference on Nam intake on Nam mice and RF + Nam mice. (Appendix A). There were significant atrophy and fibrosis of kidney appearance in RF mice, though these changes were reduced in RF + Nam mice correlation with Nam concentration (Appendix A). The weight of harvested organs were shown in Appendix A. The weight of kidney was reduced in RF mice; conversely, the weight reduction of kidney was attenuated by 0.6% Nam and 1.2% Nam administration. Kidney function, plasma creatine, urea nitrogen, and indoxyl sulfate were reduced in RF + Nam mice after 0.6% and 1.2% Nam supplementation (Figure 1c). In this adenine-induced CKD model, histological changes were observed in the renal tubules, while the glomeruli showed minimal alterations (Figure 1d). Thus, we evaluated the percentage of the remaining cortical tubular area using Image J analysis software. RF + Nam mice treated with 0.6% and 1.2% Nam exhibited higher remaining cortical tubular area compared to RF mice (Figure 1e). In addition, mRNA expression of collagen type 4 alpha 1 (*Col4a1*) was significantly increased in RF mice, and the mRNA expression was significantly reduced in RF + Nam mice after 0.6% and 1.2% Nam supplementation (Figure 1f). These results suggested that Nam prevented the progression of kidney disease by suppressing interstitial fibrosis and tubular atrophy. Then, to determine the mechanism of interstitial fibrosis and tubular atrophy suppression, inflammatory (*Tnf*, and *Il6*), oxidative stress (*p47phox*), and mitochondrial injury (*Pink1*) markers were evaluated (Figure 2a). The gene expressions of inflammatory markers, *Tnf* and *Il6*, were elevated in RF mice compared to Cont mice and Nam mice, and supplementation with 0.6% and 1.2% Nam reduced the levels of these markers. In addition, the level of *p47phox*, an indicator of oxidative stress, was reduced by 0.6% and 1.2% Nam administration. The level of *Pink1*, which protect against mitochondrial dysfunction during cellular stress by phosphorylating mitochondrial proteins, was reduced in RF mice compared to Cont mice, and significantly increased in RF mice with preventive 1.2% Nam supplementation. Furthermore, we performed immunohistochemistry staining of F4/80, which represents the presence of macrophage infiltration, to determine whether Nam administration reduce the macrophage infiltration (Figure 2b). F4/80 positive area was barely observed in Cont mice and Nam mice. RF mice presented a diffuse F4/80 positive area, while RF + Nam mice presented a reduction of F4/80 positive area. These results indicate that the renal fibrosis was occurred by infiltration of macrophages, and Nam administration reduced infiltration of macrophage.

### 2.2. Nam Supplementation Did Not Attenuate Advanced Kidney Disease

To determine the effects of Nam when administered at advanced stages of kidney disease, the mice were fed an adenine-enriched diet for 6 weeks, and then supplied with tap water with or without Nam for 4 weeks (Figure 3a), under the conditions described for previous experiments (Figure 1a). Notably, 1.2% Nam was not tolerated by RF + Nam mice, which exhibited an immediate decline in body weight. Moreover, food and water consumption dropped on day 3 of Nam administration. For ethical reasons, these mice were euthanized on day 3.

Nam mice (0.3% and 0.6% Nam) gained weight after restoration of the normal diet. In contrast, RF + Nam mice did not show changes in body weight. In both subgroups of Nam mice (0.3% and 0.6% Nam), the weight gain was proportional to food consumption, while water intake was not affected. The Nam intake in RF + Nam mice were elevated more than twice of that of Nam mice in both subgroups (0.3% and 0.6%). At week 10, the body weight was smaller in RF + Nam than in RF mice at both Nam doses (Figure 3b). These weight reductions were also seen in the harvested organs including kidneys (Appendix A). In addition, the elevation of plasma creatinine and urea nitrogen was not attenuated in RF + Nam mice after treatment with both 0.3% and 0.6% Nam (Figure 3c).

Interestingly, although kidney function was not improved by Nam administration, the plasma level of indoxyl sulfate was reduced in RF + Nam mice treated with 0.3% Nam. Nam administration did not affect adenine diet-induced macro changes (Appendix A), histological changes (Figure 3d) and the reduction of tubular area observed in RF mice (Figure 3e). In addition, mRNA expression of *Col4a1* was significantly increased in RF mice, but the mRNA expression was not reduced in RF + Nam mice (Figure 3f). Thus, Nam administration in advanced stages of kidney disease failed to restore renal function or to attenuate kidney fibrosis. Moreover, the expression of inflammatory markers and *Pink1* was not affected by either 0.3% or 0.6% Nam administration (Figure 4), suggesting that these treatments did not reduce the inflammation associated with advanced kidney injury.

### 2.3. Nam Administration Affected the Accumulation of Glycolysis, Pentose Phosphate Shuttle, and Krebs Cycle Metabolites at Early, But Not Advanced, Stages of Kidney Disease

Glycolysis and Krebs cycle supply energy for many metabolic pathways, and the pentose phosphate shuttle is essential for detoxification. The levels of representative metabolites of these three pathways were measured in the kidney using gas chromatography/mass spectrometry (GC-MS) (Figure 5a). Nam did not affect the levels of these metabolites at any of the tested concentrations. RF mice exhibited a strong accumulation of glucose 6-phosphate (G6P), fructose 6-phosphate (F6P), citric acid, and isocitric acid, as well as a pronounced decrease in the levels of 3-phosphoglyceric acid (3PG) and 2-phosphoglyceric acid (2PG) compared to both Cont and Nam mice. The prophylactic treatment with Nam at all concentrations caused comparable reductions in the levels of G6P, citrate, and isocitrate in RF + Nam mice, while dose-dependent Nam-induced increases were observed in the levels of 3PG and 2PG.

The concentrations of ribulose 5-phosphate (RI5P) and ribose 5-phosphate (R5P) were decreased in RF mice. Moreover, altered concentrations of RI5P and R5P were detected in RF + Nam mice. Furthermore, the G6P/R5P ratio, which indicates activation of the pentose phosphate pathway, was decreased in RF mice, and increased in RF + Nam mice. The concentration of F6P was increased in both RF and RF + Nam mice. However, Nam administration at advanced disease stages did not produce significant changes in the above-mentioned metabolites. 

A representative scheme of the relevant pathways showing the effects of early 0.6% Nam treatment on the concentrations of metabolites in the different experimental groups is illustrated in Figure 5B. In RF mice, the conversion of isocitric acid to 2-ketoglutaric acid (2-KG) was reduced by the treatment, as isocitric acid was increased and 2-KG decreased. The opposite result was observed in RF + Nam mice. Moreover, the malic acid-aspartate ratio (Malic Acid/Asp), which is a surrogate marker of NADH/NAD ratio, was increased in RF mice and reduced in RF + Nam mice. These results suggested that the NAD^+^ level was reduced in RF mice and restored by Nam treatment. Therefore, NAD^+^ might act as a rate-limiting co-enzyme of the glycolysis and Krebs cycle.

### 2.4. The Levels of NAD^+^ Increased in the Kidney with the Concentration of Nam

NAD^+^ synthesis from Nam mainly occurs by the salvage pathway (Figure 6a). To determine whether the NAD^+^ supply was increased by Nam supplementation, we measured the concentration of nicotinamide and NAD^+^ in the plasma and kidney, respectively, in mice subjected to prophylactic Nam treatment (Figure 6b). In Nam mice, Nam concentration in the plasma was dose-dependently elevated by Nam supplementation. Conversely, RF + Nam mice exhibited plasma Nam accumulation at low dosages but a decline at high dosages. In Nam and RF + Nam mice, intra-renal NAD^+^ concentration increased with the dosage of Nam. At a Nam dose of 0.6%, the intra-renal NAD^+^ concentration was restored to that of Cont group level. Interestingly, at a Nam dose of 1.2%, the intra-renal NAD^+^ concentration of RF + Nam mice exceeded that of control mice.

Furthermore, the mRNA expression of nicotinamide phosphoribosyltransferase (*Nampt*) and nicotinamide mononucleotide adenine transferase (*Nmnat1*) was examined. *Nampt* and *Nmnat1* expression was reduced in RF mice and restored in RF + Nam mice treated with 0.6% Nam and 1.2% Nam. No differences in *Nampt* and *Nmnat1* expression were observed between Cont and Nam mice. These data suggested that *Nampt* and *Nmnat1* had no influence on kidney NAD^+^ concentration, which was solely affected by Nam concentration.

## 3. Discussion

In this study, we demonstrated that the prophylactic administration of Nam to a mouse model of adenine-induced CKD prevented the progression of kidney disease. In Nam-administrated mice, kidney inflammation and oxidative stress were reduced by the enhanced activation of pathways of energy metabolism. In particular, the increased availability of NAD^+^ significantly activated these metabolic pathways. However, Nam administration to mice with advanced kidney disease failed to restore the renal function or ameliorate the uremic state.

### 3.1. Nam Administration May Be Beneficial in Early Stages of Kidney Disease

Several studies have shown that the supplementation of NAD^+^ or its metabolites reduces the progression of acute kidney disease. Increased NAD^+^ levels have been reported to activate sirtuin, a group of NAD^+^-consuming enzymes, and to reduce inflammation and reactive oxygen species [20]. The stimulation of NAD^+^ synthesis was found to produce renoprotective effects in the AKI model [21] and the diabetic kidney disease model [27]. However, few studies have addressed the effects of Nam in CKD models. A previous study on unilateral ureteral obstruction (UU’O) has shown that Nam administration before UU’O surgery prevents renal fibrosis by reducing the inflammation caused by IL-1 beta [23], which supports our results that continuous Nam administration prevents inflammation and fibrosis of kidney. Another study based on an advanced adenine-induced CKD rat model with feeding phosphate rich diet while administrating Nam demonstrated that Nam supplementation reduces phosphate accumulation and reduces the elevation of serum creatinine and blood urea nitrogen [28]. Another study based on an adenine-induced CKD rat model demonstrated that Nam supplementation reduces phosphate accumulation and improves creatinine clearance [28]. Our study showed that prophylactic Nam administration reduced the progression of kidney inflammation and fibrosis, thus preventing the increase of plasma creatine and urea nitrogen. In addition, Nam renoprotective effects were dose-dependent. Interestingly, the plasma concentration of Nam was elevated following the administration of low dosages of Nam and normalized at higher dosages. These results suggested that early kidney disease is associated with an increased demand for NAD^+^ and, therefore, with accelerated Nam consumption. Conversely, low-dosage Nam administration may not meet the demand for increased NAD^+^ production. Under the latter conditions, NAD^+^ supply would not prevent the decline of kidney function and, as a result, non-utilized Nam accumulates in plasma. Nam is also enrolled as uremic toxins [29] since its end products reported to exhibit deterioration of kidney function [30]. Altogether, the Nam dosages must be carefully considered to receive the therapeutic effects.

### 3.2. Nam Administration Did Not Exert Beneficial Effects in Mice with Advanced Kidney Disease

In most previous studies investigating the effects of Nam or NAD^+^ derivatives in animal models of kidney disease, the treatments were either administered at early stages of disease or prophylactically to prevent the disease onset. We also found that the effects of Nam were also explored in advanced disease settings. In our model of progressive kidney disease, the administration of a high Nam dose in drinking water was not tolerated. Notably, high doses of Nam may be associated with several adverse effects in advanced kidney disease. In our study, the intake of drinking water in RF + Nam mice was already increased at the time of starting Nam administration. Thus, a group of mice assumed high amounts of Nam immediately after the intervention. In particular, on the first day following intervention, mice with advanced kidney disease treated with 0.6% Nam assumed over 2500 mg/kg/day nicotinamide (data not shown), while those administered with 1.2% Nam assumed more than 4000 mg/kg/day nicotinamide [31]. This suggests that Nam intake in the 1.2% treatment group exceeded the tolerable dosage. The NAD^+^ synthesis occurs at every tissue, therefore the NAD^+^ which used for renal tubules are produced at renal tubule cells. Thus, the damaged renal tubule cells could not produce NAD^+^ to meet the demand. The synthesis of NAD^+^ may be reduced compared to the RF + Nam mice administered same 1.2% Nam in early CKD stage. This may be one of the reasons that administration of Nam was not effective in advanced diseases. However, our results suggest that 0.3% Nam administration reduced the accumulation of indoxyl sulfate. Therefore, the effects of lower Nam dosages should also be tested to fully establish the efficacy of this compound in advanced disease.

### 3.3. Nam Supplementation Altered the Levels of Glycolysis, Pentose Phosphate Pathway, and Krebs Cycle Metabolites in the Kidney

Previous studies have shown that diabetic kidney disease [32] and non-diabetic kidney disease [33,34] are associated with changes in the levels of glycolysis and Krebs cycle metabolites.

Glycolysis and Krebs cycle is the major pathway of ATP production, thus, associates with multiple organ dysfunction. The impairment of ATP production is crucial for proximal renal tubule, since it consumes large amount of ATP for absorption of electrolytes, and molecules [35]. Proximal renal tubule cells contain a lot of mitochondria, which is the main source of ATP production. Therefore, the prognosis of acute kidney injury is determined by the degree of damage on proximal tubules, and insufficient recovery leads to development of CKD [36]. The reduction of NAD^+^ leads to impairment of glycolysis and Krebs cycle, and supplementation of NAD^+^ rescues these metabolic disorders [37]. Indeed, the inhibition of glycolysis reduced renal fibrosis in CKD model mice [38]. This study implies that glycolysis should be maintained in a certain degree of activation.

In our study, CKD mice exhibited increased levels of glucose, G6P, F6P, citric acid, isocitric acid, fumaric acid, and malic acid, and decreased levels of 3-PG, 2-PG, and 2-KG, in line with previous reports. Interestingly, we found that Nam altered the profile of various metabolites. These changes were particularly pronounced before and after the NAD^+^-consuming events. For instance, isocitric acid conversion to 2-KG by isocitrate dehydrogenase requires NAD^+^. In RF mice isocitrate was increased and 2-KG decreased. However, in RF + Nam mice a sharp decrease in isocitric acid and an increase of 2-KG were observed. Thus, we assumed that NAD^+^ supplementation regulated the activation of the Krebs cycle. We observed similar changes in association to all NAD^+^-consuming reactions during glycolysis and the Krebs cycle (Figure 6B). Consequently, adjustments in glycolysis were involved in the preventive effects of prophylactic Nam on the progression of kidney disease.

### 3.4. Perspectives in the Use of Nam for the Treatment of CKD

In our study, we used an adenine-induced CKD model. This model is characterized by tubulointerstitial damage without substantial glomerular changes. In humans, adenine phosphoribosyltransferase (APRT) deficiency has similar consequences [39]. A recent study has shown that the excess of 2,8-dihydroxyadenine (2,8-DHA) in the urine results in the accumulation of an insoluble fraction of this compound, which forms crystals in the renal tubule, causing obstructive nephropathy. Moreover, the 2,8-DHA crystals adhere to the renal tubules, while renal epithelial cells surround the crystals and transport them to the interstitial lesion to form granulomas [40]. During this process, TNF receptors play a critical role in adhesion, as observed in oxalate nephropathy [41]. In our study, Nam administration reduced *Tnf* and *Il6* expression, which likely resulted in the downregulation of TNF receptors. Nam might also be effective in the treatment of inflammation associated with other renal diseases, such as diabetic kidney disease [42] and IgA nephropathy [43], which also involve TNF receptors.

A comprehensive scheme of our study is shown in Figure 7. Excess adenine intake induces inflammation and increases oxidative stress. These changes may lead to a high demand for energy supply. The glycolysis and the Krebs cycle promotes the production of ATPs to meet the demand. A high content of Nam in the diet would result in increased NAD^+^ availability, which in turn may regulate glycolysis and Krebs cycle. Thus, the maintenance of energy metabolism would ultimately result in a renoprotective effect.

Our study has several limitations. We only used a single animal model of CKD, the adenine-induced model, in which primary renal injury is detectable in tubular and interstitial areas. Therefore, whether similar results would be obtained with glomerular disease remains to be determined. Moreover, since a time-course analysis was not performed, it remains unclear whether Nam may be effective in early disease stages. In addition, we did not test our results in human patients.

## 4. Conclusions

In conclusion, we have demonstrated that Nam supplementation to mice with adenine-induced CKD reduced the degree of inflammation, oxidative stress, and renal fibrosis, and promoted energy metabolism, thus preserving the renal function.

## 5. Materials and Methods

### 5.1. Animal Experiments

Male C57BL/6 mice of 5–7 weeks of age were purchased from CLEA Japan (Tokyo, Japan). The mice were kept under a 12 h day/night cycle and fed a normal diet (Oriental Yeast. Co., Ltd., Tokyo, Japan). Tap water was supplied for one week prior to the experiments. At seven weeks of age, the mice were randomly divided into control and adenine groups and co-housed. At eight weeks of age, the control group was maintained under normal diet, while the adenine group was fed a diet containing 0.2% adenine (FUJIFILM Wako Pure Chemicals Corp., Tokyo, Japan) for six weeks. For the prophylactic treatments, the mice were supplied with tap water containing 0.3%, 0.6% or 1.2% Nam (Sigma-Aldrich, St. Louis, MO, USA) or control tap water ad libitum. For advanced-disease treatments, the mice were supplied with 0.3 or 0.6% Nam or control water ad libitum for four consecutive weeks from week 15. At the end of this period, urine and blood samples were collected. The mice were anesthetized with 3% isoflurane (FUJIFILM Wako Pure Chemicals Corp., Tokyo, Japan) and then euthanized by cervical dislocation. Mouse organs were collected and quickly preserved in liquid N_2_ or fixed in 2% paraformaldehyde for analysis. All animal experiments were approved by the Animal Committee of Tohoku University School of Medicine (Approval No. 2019Pha-010, 15 February 2019). The experimental protocols and animal care were performed according to the guidelines for the care and use of animals established by Tohoku University.

### 5.2. Histological Analysis

The mouse kidneys were fixed in 2% paraformaldehyde (FUJIFILM Wako Pure Chemicals Corp., Tokyo, Japan) and embedded in paraffin. Kidney sections were stained with hematoxylin and eosin, periodic acid–Schiff, and combined Verhoeff and Masson trichrome (E-M) stain, immunohistochemistry stain for F4/80. Stained sections were imaged using an BZ9000 microscope (Keyence, Tokyo, Japan). For analysis of the interstitial fibrosis and tubular atrophy area, the percentage of the tubular area in the whole kidney was evaluated in E-M-stained kidney sections using the National Institutes of Health Image J analysis software.

Immunohistochemistry for F4/80 were performed as previously described [44]. We used rat-anti-mouse F4/80 monoclonal antibody (1:200, Clone A3-1, Bio-Rad, Hercules, CA, USA) as the primary antibody. Negative controls remained negative (data not shown). The ratio of the F4/80 positive area to tubulointerstitial area was assessed using Image J.

### 5.3. Quantitative Real-Time PCR Analysis

Whole kidneys were homogenized in TRI-Reagent (Molecular Research Center, Inc. Cincinnati, OH, USA) and extracted according to the manufacturer’s instructions. cDNA synthesis was performed using the iScript cDNA synthesis kit (Bio-Rad, Hercules, CA, USA). The primers were purchased from Takara (Kusatsu, Japan), and their set ID were *Nampt*: MA166767, *Nmnat1*: MA111699, *Hprt*: MA031262, and *p47phox*: MA175153. The sequences of *Col4a1* and *Pink1* were Forward: GGCTCTGGCTGTGGAAAA and Reverse: CCAGGTTCTCCAGCATCACC, and Forward: GCACGCTTAGCTGCAAATGT and Reverse: AGCCAGGCGATCATCTTGTC, respectively. Gene expression was measured with SYBR Premix Taq II (Takara, Kusatsu, Japan) and Bio-Rad CFX real-time PCR systems. As a reference gene, hypoxanthine-guanine phosphoribosyl transferase (*Hprt*) was used.

### 5.4. LC-MS/MS Measurements for Indoxyl Sulfate and Creatinine

This measurement reference to the previous our reported methods [45]. For sample preparation, 150 μL of 0.1% formate methanol (containing 2.0 µg/mL creatinine-d_3_ and 1.25 µg/mL indoxyl sulfate-d_4_) were added to 50 μL of each plasma and vortexed for 1 s. The samples were then sonicated for 5 min and centrifuged at 16,400× *g* for 20 min at 4 °C. The supernatant was filtered through membranes (pore size: 0.22 μm; Merck Millipore, Billerica, MA, USA). Quantitative analysis of indoxyl sulfate was performed using LC-MS/MS using a Prominence LC system (Shimadzu, Kyoto, Japan) coupled to a TSQ Quantiva mass spectrometer (Thermo Fisher Scientific, Waltham, MA, USA), and operated in negative mode. Each sample (5 µL) was injected onto a 150 × 2.0 mm YMC-Pack Pro C18, 3-µm column (YMC, Kyoto, Japan) at a flow rate of 0.3 mL/min, as we previously described [16]. For gradient elution, mobile phase A was 10 mM ammonium acetate, and mobile phase B was acetonitrile. Linear and stepwise gradients were programmed as follows: 0–1 min: 0–10% solvent B; 1–2 min: 10–40% solvent B; 2–3 min: 40–80% solvent B; 3–5 min: 80–100% solvent B; 5–7 min: 100% solvent B; 7–10 min: 0% solvent B. Quantification analyses by MS/MS were performed by a selected reaction monitoring mode (SRM), in which the transitions of the precursor ion to the product ion and collision energy (eV) were monitored: *m*/*z* 212→80, 21 eV for indoxyl sulfate; *m*/*z* 216→80, 30 eV for indoxyl sulfate-d_4_. Quantitative analysis of creatinine was performed using LC-MS/MS, and operated in the positive mode. Each sample (5 µL) was injected onto a 150 × 2.0 mm YMC-Pack Pro C18, 3-µm column (YMC, Kyoto, Japan) at a flow rate of 0.2 mL/min. For gradient elution, mobile phase A was 0.1% formate in pure water, and mobile phase B was acetonitrile. Linear and stepwise gradients were programmed as follows: 0–1 min: 2–5% solvent B; 1–3 min: 5 –20% solvent B; 3–5 min: 20–50% solvent B; 5.1–7 min: 100% solvent B; 7–10 min: 2% solvent B. Quantification analyses by MS/MS were performed by a SRM mode, in which the transitions of the precursor ion to the product ion and collision energy (eV) were monitored: *m*/*z* 114→86, 11 eV for creatinine; *m*/*z* 117→89, 12 eV for creatinine-d_3_. Spray voltage was 3000 V, vaporizer temperature was 350 °C, and ion transfer tube temperature 350 °C.

### 5.5. Sample Preparation for GC-MS Measurements

The methods here reference our previous reported methods [46]. We prepared a homogenization buffer containing 2 mM Tris-HCl, 1 mM EDTA, 0.25 M sucrose, and one tablet of cOmplete mini (Sigma-Aldrich, St. Louis, MO, USA). The kidney lysates was prepared by homogenizing 15–25 mg of kidney sample in 3 µL of homogenizing buffer per 1 mg of tissue with a Digital Homogenizer HK-1 (ASONE, Tokyo, Japan). Fifty milliliters of lysate were added to 250 µL of a solution containing 55 % methanol and 22% chloroform dissolved in distilled water. Then, 0.5 mg/mL 2-isopropylmalate (10 μL; Sigma-Aldrich) dissolved in distilled water was added and incubated in Thermo mixier C (Eppendorf) at 37 °C with 1200 rpm shaking for 30 min. The samples were centrifuged at 4 °C and 16,000× *g* for 3 min. The supernatant was collected and added to 200 L of distilled water, and the samples were centrifuged at 4 °C and 16,000× *g* for 3 min. The supernatant was collected and added to 10 μL of 1 μg/mL citrate-d4. The sample was dried using an evaporator under reduced pressure. For oximation, 20 mg/mL methoxyamine hydrochloride (80 μL; Sigma-Aldrich) dissolved in pyridine were mixed with the lyophilized sample, sonicated for 20 min, and shaken at 1200 rpm for 90 min at 30 °C. Next, 40 μL of N-methyl-N-trimethylsilyl-trifluoroacetamide (MSTFA) (GL Sciences, Tokyo, Japan) were added for derivatization. The mixture was then mixed at 1200 rpm for 30 min at 37 °C, centrifuged at 16,000× *g* for 5 min at 4 °C, and the resulting supernatant (1 μL) was subjected to GC-MS.

### 5.6. GC-MS Measurements

This measurement reference to the previous our reported methods [46]. GC-MS analysis was performed using a GC-MS QP2010 Ultra spectrometer (Shimadzu) with a fused silica capillary column (BPX-5; 30 m × 0.25 mm inner diameter, film thickness: 0.25 μm; Shimadzu) and a front inlet temperature of 25 °C, and a helium gas flow rate through the column of 39.0 cm/s. The column temperature was held at 60 °C for 2 min, then raised by 15 °C/min to 330 °C and maintained for 3 min. The interface and ion source temperatures were 280 and 200 °C, respectively. To perform a semi-quantitative assessment, the peak height of each quantified ion was calculated and normalized using citrate-d4 and 2-isopropylmalate peak height and protein concentration. Protein concentrations were determined with Bio-Rad Quick Start Bradford 1x Dye Reagent according to the manufacturer’s instructions. The retention times and SRM conditions for the derivatized metabolites are summarized in Appendix A.

### 5.7. NAD^+^ Measurements

The NAD^+^ concentration in the kidney was measured using the NAD/NADH Assay Kit-WST (Dojindo Laboratories, Kumamoto, Japan) according to the manufacturer’s instructions. Briefly, 10 mg of frozen kidney samples were homogenized in extraction buffer (20 µL/µg of tissue) using a Digital homogenizer HK-1 (ASONE, Tokyo, Japan). The lysate was filtrated with an MWCO 10K filtration tube (PALL, New York, NY, USA) by centrifuging at 4 °C and 12,000× *g* for 15 min. The filtrated solution was divided into two portions. The first portion for measurement of total NAD^+^ and NADH was diluted with control buffer and kept on ice. The second portion was heated at 60 °C for 1 h to inactivate NAD^+^ and measure NADH. The samples were placed in 96-well microplates, and a mixture of assay buffer and enzyme was added to each well to start the reaction. After incubating at 37 °C for 40 min, the absorbance at 595 nm was measured by a microplate reader (SPECTRA MAX 190, Molecular Devices, San Jose, CA, USA). The NAD^+^ concentration was calculated by subtracting the NADH concentration from the total NAD^+^ and NADH concentrations.

### 5.8. BUN Measurements

Plasma blood urea nitrogen (BUN) was measured using a BUN colorimetric detection kit (ARBOR ASSAYS, Ann Arbor, MI, USA) according to the manufacturer’s instructions.

### 5.9. Statistical Analysis

JMP Pro software version 14.2.0 (SAS Institute Inc., Cary, NC, USA) was used for statistical analysis. Differences were considered statistically significant at *p* < 0.05. Statistical comparisons of multiple groups were made using ANOVA and the Steel–Dwass test.

## Figures and Tables

**Figure 1 toxins-13-00050-f001:**
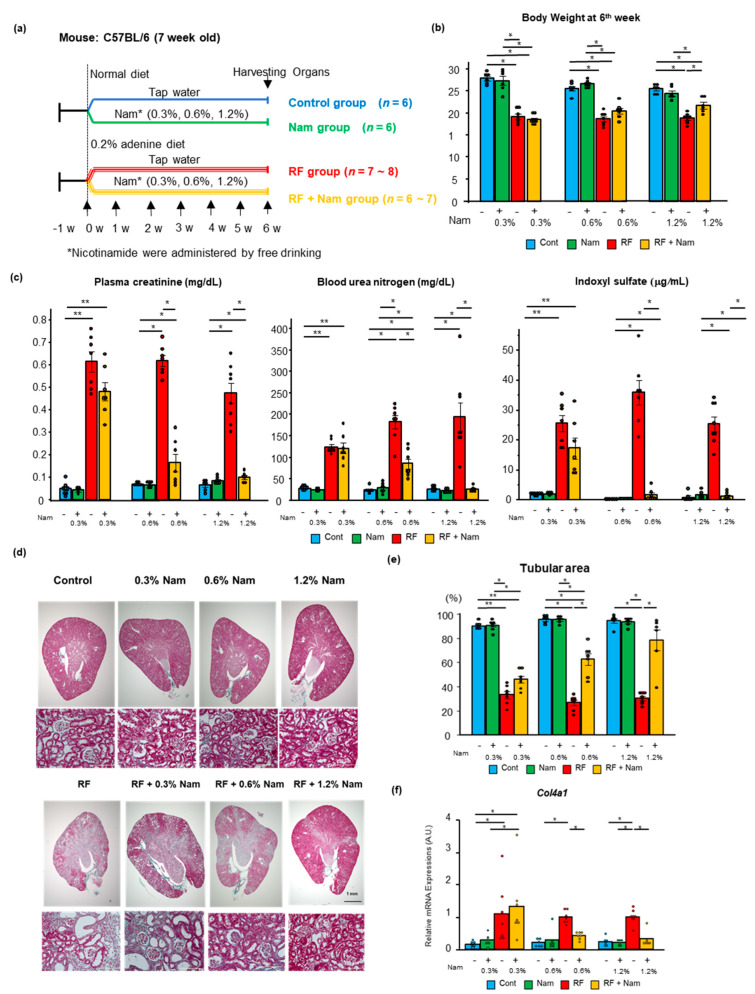
Prophylactic Nam supplementation prevented the progression of kidney disease. (**a**) Experimental protocol. C57BL/6 mice were fed a normal diet or a 0.2% adenine-containing diet. Nam was dissolved in tap water at three different concentrations: 0.3%, 0.6%, and 1.2%. (**b**) Body weight of mice at the end of the study (6th week). * Steel–Dwass test, *p* < 0.05. (**c**) Plasma concentrations of creatinine, urea nitrogen, and indoxyl sulfate. Steel–Dwass test * *p* < 0.05, ** *p* < 0.01. (**d**) Representative images of Elastica Masson Goldner-stained kidney sections. (**e**) Quantitative analysis of remaining cortical tubular area by Image J software. (**f**) The mRNA expression of collagen type IV alpha 1 (*Col4a1*) in the kidney. Steel Dwass test. * *p* < 0.05.

**Figure 2 toxins-13-00050-f002:**
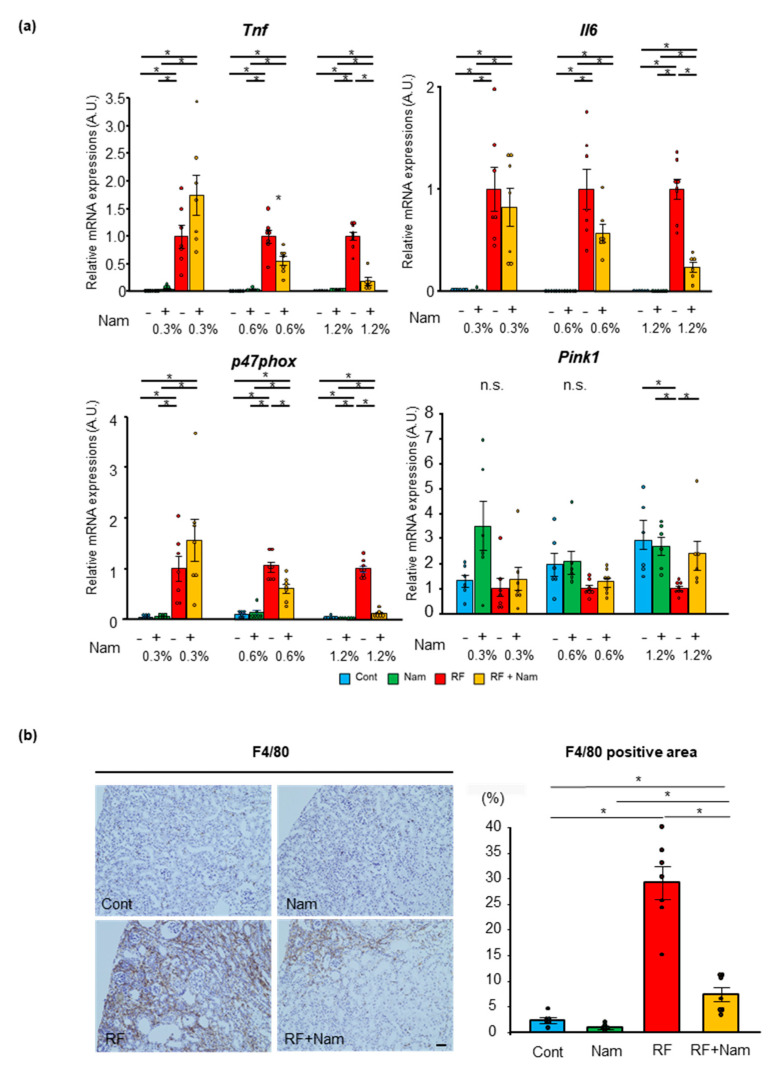
Kidney inflammation, oxidative stress mitochondrial injury and macrophage infiltration. (**a**) Quantitative real-time PCR analysis showed that kidney inflammation and oxidative stress were reduced, and mitochondrial injury was recovered following prophylactic administration of Nam. All mRNAs were normalized to *Hprt* expression. (**b**) Representative images of immunohistochemistry staining for F4/80 and quantitative analysis of F4/80 positive area performed by ImageJ software on 0.6% Nam administration experiments. Scale bar shows 40 µm. Steel–Dwass test. * *p* < 0.05.

**Figure 3 toxins-13-00050-f003:**
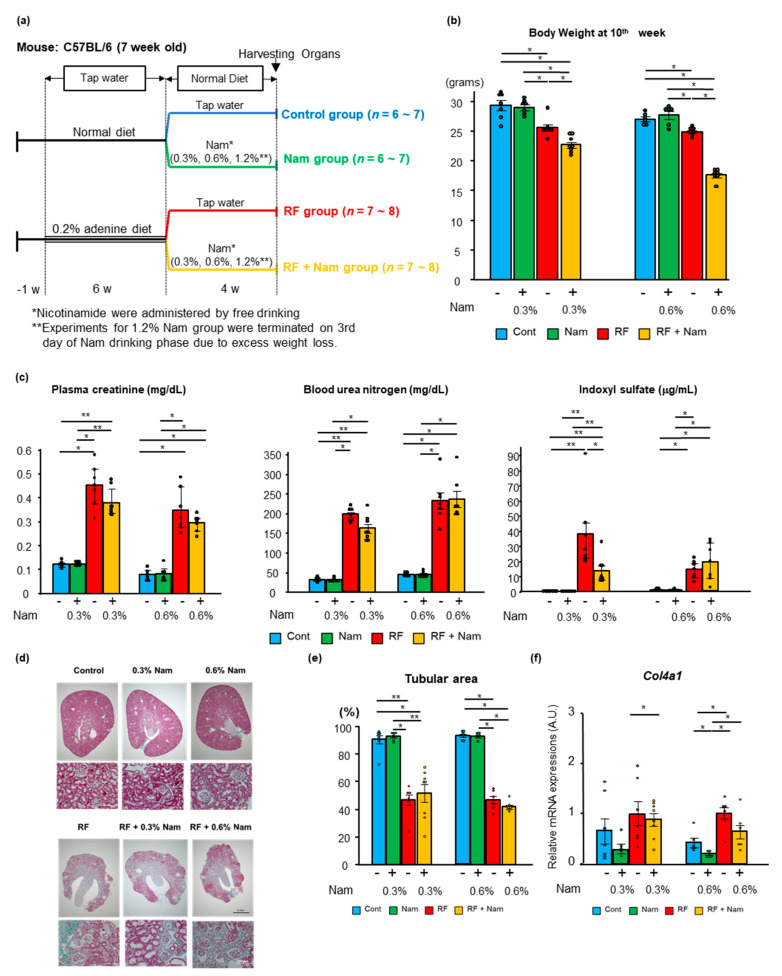
Nam administration in advanced disease stages did not improve renal fibrosis and function. (**a**) Experimental protocol. First, a normal or a 0.2% adenine-containing diet was supplied to C57BL/6 mice for six weeks. Then, a normal diet was supplied to all animals, and half of the mice in each group were administered with Nam dissolved in tap drinking water for four weeks. Three concentrations of Nam were used: 0.3%, 0.6%, and 1.2%. (**b**) Body weight of the mice at the end of the study (6th week). * Steel–Dwass test, *p* < 0.05. (**c**) Plasma concentrations of creatinine, urea nitrogen, and indoxyl sulfate. Steel–Dwass test * *p* < 0.05, ** *p* < 0.01. (**d**) Representative images of Elastica Masson Goldner-stained kidney sections. (**e**) Quantitative analysis of remaining cortical tubular area using Image J software. (**f**) The mRNA expression of collagen type IV alpha 1 (Col4a1) in the kidney. Steel–Dwass test * *p* < 0.05.

**Figure 4 toxins-13-00050-f004:**
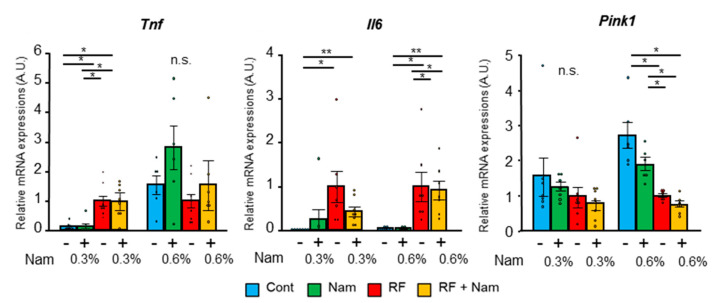
Nam administration to mice with advanced kidney disease did not reduce kidney inflammation or oxidative stress. Quantitative real-time PCR was performed from renal mRNAs. All mRNAs were normalized to *Hprt* expression. Steel–Dwass test * *p* < 0.05, ** *p* < 0.01, n.s.: no significance.

**Figure 5 toxins-13-00050-f005:**
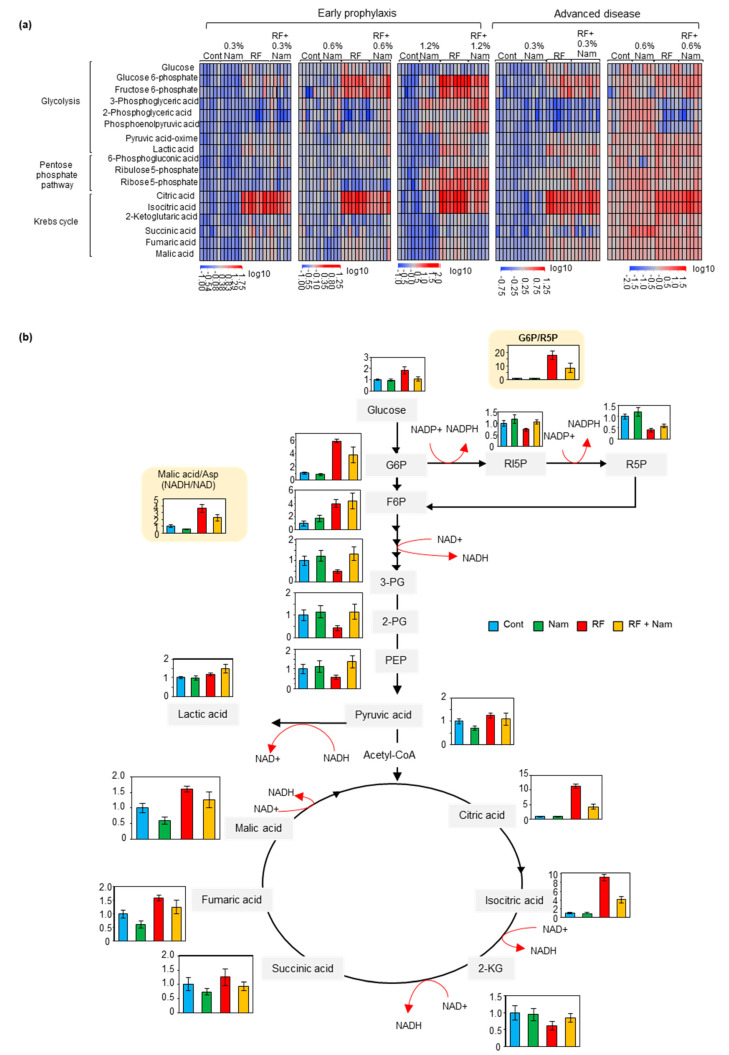
Metabolomics analysis showed that Nam modified the profile of glycolysis, pentose phosphate pathway, and Krebs cycle metabolites. (**a**) Heatmap of kidney metabolites in each experimental group. (**b**) Graphical representation of glycolysis, pentose phosphate pathway, and Krebs cycle and comparison between the levels of the relevant metabolites. Abbreviations: G6P, glucose 6-phosphate; F6P, fructose 6-phosphate; 3-PG, 3-phosphoglyceric acid; 2-PG, 2-phosphoglyceric acid; RI5P, ribulose 5-phosphate; R5P, ribose 5-phosphate; 2-KG, 2-ketogulutaric acid; PPP, pentose phosphate pathway.

**Figure 6 toxins-13-00050-f006:**
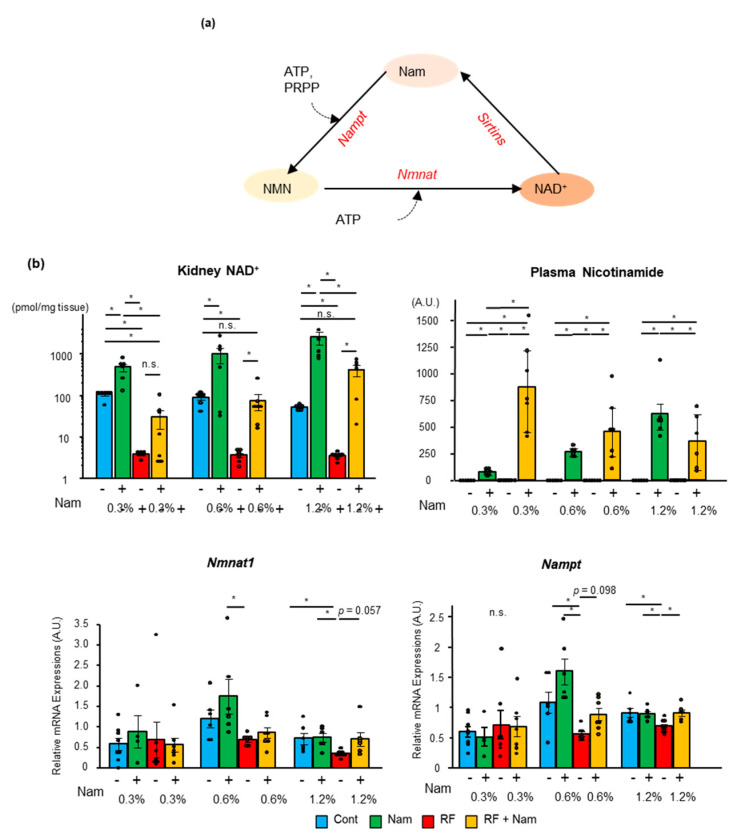
NAD^+^ concentration increased in the kidneys following Nam administration. (**a**) A brief scheme of the salvage pathway in NAD^+^ metabolism. (**b**) Kidney NAD^+^ concentrations, plasma Nam concentrations, and relative expression of *Nampt* and *Nmnat1* in the kidneys. mRNA expression was normalized to that of *Hprt*. Steel Dwass test. * *p* < 0.05, n.s.; no significance.

**Figure 7 toxins-13-00050-f007:**
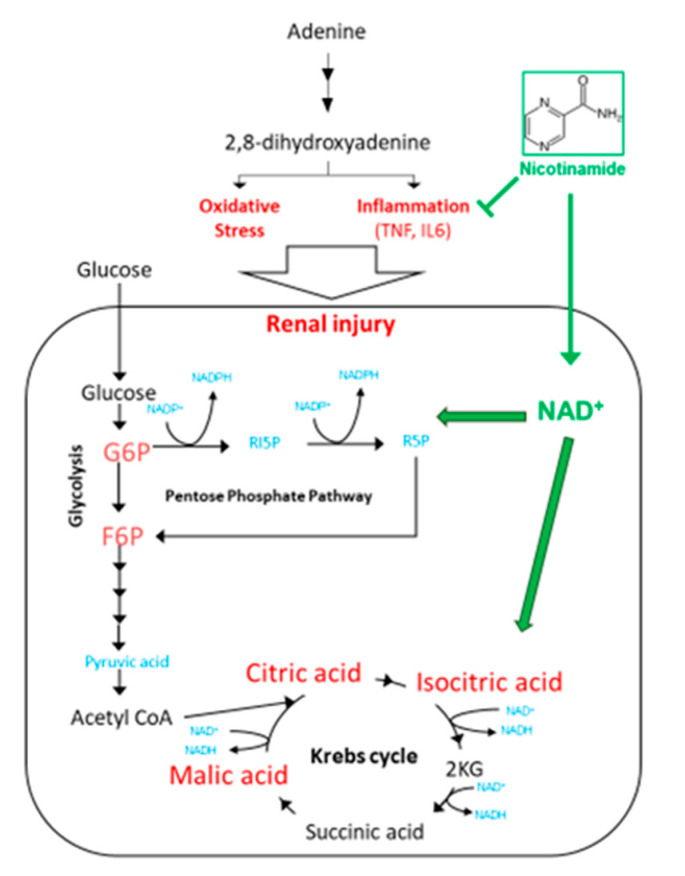
Proposed mechanism of Nam-induced renoprotection based on our findings.

## Data Availability

Data are available upon request, please contact the contributing authors.

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
