# Peer review of "Nicotinamide Attenuates the Progression of Renal Failure in a Mouse Model of Adenine-Induced Chronic Kidney Disease"

_toxins, 2021, doi:10.3390/toxins13010050_

Round 1
Reviewer 1 Report
In the present study, it described a beneficial effect of Nicotinamide (Nam) in murine model of Adenine induced-CKD. This effect was observed with prophylactic treatment of Nam, but not with Nam administration in advanced stage of CKD. The manuscript is concise and clear, although, there are several suggestions that should be considered to improve the quality of the manuscript.
- Renal Nam protection in AKI has been also observed by others group, indeed it has been tested in cardiac surgery patients, this would be cited in the introduction.
- In the image of Masson, magnification is too low to get a “good feel”, the authors should show images with more detail. What means non-interstitial fibrosis, how has it been quantified?
- Figure 1-E, the RF group of 1.2 % Nam showed less injury that the same group in 0.3% Nam, they should present similar levels of injury? The same in Figure 2, 4, 5 and 6, RF groups without Nam present different levels between them, maybe I do not understand the graphic, could the authors clear this point?
- The authors should to perform a more specific experiment to analyzed fibrosis, as quantification of Collagen or alpha-SMA expression
- To clearly determinate the Nam effect over inflammation the infiltrate of macrophage should be quantified, e.g. immunohistochemistry of F4/80.
- Mitochondria is the mainly ATP source, could the authors analyze mitochondrial injury in Adenine-CKD and the effect of Nam?
Author Response
Thank you very much for reviewing our manuscript and constructive comments. Attached please find our revised manuscript entitled “Nicotinamide Attenuates the Progression of Renal Failure in a Mouse Model of Adenine-induced Chronic Kidney Disease”.

Reviewer 2 Report
They suggested that NADm administration could be protective of CKD prevention. There is still much to be improved or revised.
1. Line 28. Which homeostasis do they mean?
2. Figure 1-6
The control and RF groups did not take Nam, but all the pictures have the concentration of the Nam as the title. Consider better notation for the pictures to reduce confusion for the readers.
3. Results 2.1
Description for body weight and intake is too wordy. Condense the descriptions, especially of supp data.
4. Figure 1 & 3
IFTA is more intuitive than non-IFTA
They should show histology with higher magnification. It is difficult to differentiate the injuries of glomeruli and tubules with the present photos.
5. Results 2.2
This part is a kind of negative result. They should delete some supp data for weight and intake. Too many 'moreover' over the part.
6. Results 2.3
Divide paragraph according to content. It is not easy to follow the sentences.
7. Line 174-177
Do they describe NAD in the kidney in the sentences?
8. Line 205
Describe the difference with the previous similar study.
9. Line 213
How do they conclude about non-utilized NAD?
10. Line 226
Compare the amount of NAD in the early CKD group with 1.2% Nam.
11. Line 240-244, 261-263
What is the relationship between the promotion of the Krebs cycle and progression to the CKD. There was no explanation between energy demand and renoprotection.
12. Line 365
Describe the measurement method of Cr and inulin
Author Response

(The authors gave the same response as above.)

Round 2
Reviewer 1 Report
The authors have performed a variety of additional experiments to strenghten the manuscript. All concerns were carefully addressed.
Reviewer 2 Report
They completely responded to all comments